# Soil Nitrogen Dynamics in a Managed Temperate Grassland Under Changed Climatic Conditions

Mona Giraud [1,*], Jannis Groh [1,2], Horst H. Gerke [2], Nicolas Brüggemann [1], Harry Vereecken [1] and Thomas Pütz [1]

[1] Agrosphere (IBG-3), Institute of Bio- and Geosciences, Forschungszentrum Jülich, 52425 Jülich, Germany; j.groh@fz-juelich.de (J.G.); n.brueggemann@fz-juelich.de (N.B.); h.vereecken@fz-juelich.de (H.V.); t.puetz@fz-juelich.de (T.P.)

[2] Leibniz Centre for Agricultural Landscape Research (ZALF), Research Area 1 "Landscape Functioning", Working Group "Hydropedology", 15374 Müncheberg, Germany; gerke@zalf.de

\* Correspondence: m.giraud@fz-juelich.de

**Abstract:** Grasslands are one of the most common biomes in the world with a wide range of ecosystem services. Nevertheless, quantitative data on the change in nitrogen dynamics in extensively managed temperate grasslands caused by a shift from energy- to water-limited climatic conditions have not yet been reported. In this study, we experimentally studied this shift by translocating undisturbed soil monoliths from an energy-limited site (Rollesbroich) to a water-limited site (Selhausen). The soil monoliths were contained in weighable lysimeters and monitored for their water and nitrogen balance in the period between 2012 and 2018. At the water-limited site (Selhausen), annual plant nitrogen uptake decreased due to water stress compared to the energy-limited site (Rollesbroich), while nitrogen uptake was higher at the beginning of the growing period. Possibly because of this lower plant uptake, the lysimeters at the water-limited site showed an increased inorganic nitrogen concentration in the soil solution, indicating a higher net mineralization rate. The $N_2O$ gas emissions and nitrogen leaching remained low at both sites. Our findings suggest that in the short term, fertilizer should consequently be applied early in the growing period to increase nitrogen uptake and decrease nitrogen losses. Moreover, a shift from energy-limited to water-limited conditions will have a limited effect on gaseous nitrogen emissions and nitrate concentrations in the groundwater in the grassland type of this study because higher nitrogen concentrations are (over-) compensated by lower leaching rates.

**Keywords:** nitrogen cycle; weighable lysimeter; nitrate leaching; nitrous oxide emission; plant nitrogen uptake; extensive managed temperate grassland; Budyko

## 1. Introduction

Natural grasslands provide a wide range of ecosystem services, like regulation of climate and nutrient cycling as well as fodder production [1]. One critical aspect of the preservation of the grasslands' ecosystem services is the regulation of the nitrogen (N) budget in the soil [2]: A change in the amount or form of inorganic N in a grassland can, for instance, decrease the plants' nitrogen uptake (N_PU) rate [3]; and an excess in soil N can lead to N-loss [4] and a decrease in biodiversity [5].

The management of the N-budget also affects the global ecosystem: Grasslands can be sources of N via the release of nitrous oxide ($N_2O$), ammonia ($NH_3$), and nitrate ($NO_3^-$) [6]. An increase in reactive N in the atmosphere leads to the acidification of wet deposition (WD) [7], the destruction of the ozone layer in the stratosphere, and contributes to global warming [8,9]. Increase in surface water nitrogen concentration can lead to a change in biodiversity, eutrophication, and acidification [7,9].

As explained by Collins et al. [10] the global climate is already experiencing severe changes, like an increase in air temperature ($T_{air}$) and variation in precipitation (Prec) frequencies and intensities. These changes will likely become more pronounced [10] and will

affect the nitrogen cycle in grasslands [11,12]. It is therefore essential to better understand the dynamics, which influence the N-fluxes, pools, and turnover processes in grassland ecosystems and their variation in the future.

An increase in $T_{air}$ and in the corresponding soil temperature leads to increasing microbial activity and rates of soil biogeochemical processes such as mineralization, nitrification, and denitrification, although increasing net rates of these processes have not always been observed in grassland soils [13–15]. A higher mineralization rate of soil organic matter can lead to an increase in inorganic N-concentration in soil solution, and thus to potentially higher N-leaching (N_Dr) [14] and gaseous $N_2O$ emissions [16,17]. Moreover, the solubility of $N_2O$ in the soil solution decreases with increasing $T_{air}$, which is also leading to higher gaseous $N_2O$ emissions [17]. On the other hand, a higher $T_{air}$ can cause a higher N_PU if it does not lead to water or heat stress [18]. Higher N_PU in response to an increasing $T_{air}$ controls soil solution chemistry [19], lowers the concentration of soil dissolved inorganic nitrogen (DIN), N_Dr [20], and gaseous emissions [21,22].

In contrast, mineralization, nitrification, and denitrification were found to be lower under drier environmental conditions [23,24]. Droughts can cause a decrease in microbial biomass N and in N_PU in grasslands [25–28] and lower soil water content, both leading to a higher concentration of DIN in the soil solution [22]. The effects of drought periods followed by rewetting on the N-cycle vary [29]. Intense drying periods can cause a decrease in cumulative N-mineralization. On the contrary, short drying of the soil—below a site-specific threshold of desiccation intensity—followed by periods where $T_{air}$ and soil moisture are near their optimal values, can lead to a significant increase, known as the Birch effect. Moderate droughts in temperate grasslands were seen to increase mineralization [29], $N_2O$ emissions [16,30,31], and N_Dr from fertilization [4], especially in case of preferential flow towards subsurface drains [32].

The effects of a drier and warmer climate on the soil N-cycle dynamics depend on the history and initial conditions of the sites and on the amplitude of the changes. For this reason, N-turnover variations are site-specific, and there is a limited correlation between the potential—laboratory—and actual—field—turnover [26,33,34]. In addition, there is a strong interrelation between the effects of $T_{air}$ and Prec on the N-cycle [35]. In spite of this, few studies have focused on the relationship between soil moisture- and temperature-driven changes of N-turnover [35]. Finally, feedback effects (e.g., shift or adaptation of the microbial and plant communities, depletion of the soil N-pools) could lead to differences between the short- and long-term responses of the grassland ecosystem to environmental changes [12,28].

Consequently, there is a need for long-term, holistic, and non-destructive field experiments in natural grasslands, in which the main soil state variables and boundary fluxes can be continuously monitored [36]. Zistl-Schlingmann et al. [34], Ineson et al. [19] and Fu et al. [14] conducted a one, two, and three year-long lysimeter study, respectively, to observe the interaction between the water and the nitrogen balances of grasslands. However, they could not compare the ecosystem balances of an energy-limited climate, where energy by reference evapotranspiration is a limiting factor for actual evapotranspiration (ETa), with those for a water-limited climate, where water is the limiting factor for ETa (see Section 2.2).

The above literature review allows us to hypothesize that a change in the grassland ecosystem from an energy- to a moderately water-limited environment with overall drier and warmer climatic conditions and altered seasonal Prec distribution—an accelerated version of the climatic shift already occurring in the northwest of Germany—will lead to a decreasing grassland N_PU since the grassland ETa will be limited by water stress [22,37]. This will contribute to an increase in net mineralization rate [22]. Moreover, the combination of fertilization and increasing dry periods will lead to a higher amount of N in the soil (potentially) available for leaching and gaseous emissions [4,31].

The aim of this study was to test the hypotheses by analyzing data from high-precision weighing grassland lysimeters with the same soil and initial vegetation cover under energy and water-limited—while both temperate—site conditions in northwestern Germany.

We also compare observations from 2018, an especially dry year in Germany [38], with observations from wetter years.

## 2. Materials and Methods

### 2.1. Experimental Setup, Sites and Soils

The experimental setup is part of the Germany-wide TERENO- observatory network (TERrestrial ENvironmental Observatories; https://www.tereno.net/ accessed 1 April 2020) that intends to provide extensive data sets from high precision non-destructive measurements of the soil–plant–atmosphere continuum [39]. The TERENO SOILCan sub-project, implements an adapted "Space for Time" (SFT) substitution approach: lysimeters are transferred across TERENO study sites, along temperature and precipitation gradients. The translocation led to an abrupt rather than gradual climate regime variation, contrary to most SFT substitution studies. Rather than following a gradual change, the aim is to compare the transferred soil with the soil at the original site. It is thus possible to account for unsuspected effects from the past [40,41]. The high-precision weighable SOILCan lysimeters are used to observe the effect of complex processes on the soil–plant–atmosphere relations including soil water and nitrogen dynamics [39]. This way, effects of climatic shifts on the soil ecosystem can be observed under actual climatic conditions for Germany. For more information on the TERENO-SOILCan network and its adapted SFT substitution approach see Pütz et al. [39] andGroh et al. [40].

The study sites are part of the Eifel/Lower Rhine Observatory, North Rhine-Westphalia, Germany—of the TERENO SOILCan Lysimeter network [39]. One site is located near Rollesbroich—50°37′12″ N 6°18′15″ E, 515 m asl, 1.63° slope, a grassland dominated by *Lolium perenne* [42], and the other near Selhausen—50°52′7″ N 6°26′58″ E, 104 m asl, 0.3° slope [43]. The annual average $T_{air}$ in Rollesbroich measured for the period 2012–2018 was 8.3 °C, while it was 11.1 °C in Selhausen. Mean annual Prec for the period 2012–2018, as measured with lysimeters, was 1060 mm year$^{-1}$ in Rollesbroich and 665 mm year$^{-1}$ in Selhausen. A graphical presentation of the TERENO network and the two study sites can be found in Figure A1 (see Appendix A).

In 2010, nine lysimeters were filled with undisturbed soil monoliths at Rollesbroich, six of them—see TERENO-portal www.tereno.net/ddp/, accessed 1 April 2020, lysimeter ID: Ro_Y_011 to Ro_Y_016—were installed in the close vicinity of the monolith excavation site in Rollesbroich, and three—lysimeter ID: Se_Y_021, Se_Y_025, and Se_Y_026—were transferred to the site Selhausen. From these lysimeters, data from 2012 to 2018 was used as explained below. Data from 2011 was not considered because the lysimeters required a period of at least one hydrological year to adapt to the sites' conditions [37].

Each lysimeter consists of an outer cylindrical stainless steel shell of 1.50 m height and 1 m$^2$ surface area. A silicon porous suction cup rake (SIC40, UMS GmbH, Munich, Germany) at the bottom of the soil monolith lysimeters at approximately 145 cm depth is connected to a bi-directional pump and a seepage water tank. This pump allows for the transport of water into or out of the lysimeter to reduce any difference in matric potentials between the surrounding field soil and the lysimeter soil at 1.4 m depth. Through this pump control, the upward- or downward-directed water flux at the lysimeter bottom boundary is supposed to mimic the real water movement in the surrounding field soil. The lysimeters and the seepage water tanks are equipped with weight cells of high resolution—0.01 kg for the lysimeters, 0.001 kg for the tanks. Aliquots of the seepage water—only downward water flux—are collected in a bottle attached to the water tank. A detailed description of the lysimeters and setup is given in Pütz et al. [39] and Groh et al. [40].

Water samples from the tanks, aliquots and deposition sampler were collected every two weeks and analyzed for DIN, $NO_3^-$, and $NH_4^+$. The DIN was measured via a TOC—total organic carbon—Analyzer (TOC-VCPH, Shimadzu Corporation, Kyoto, Japan) with a total nitrogen measuring unit (TNM-1, Shimadzu Corporation, Kyoto, Japan). The $NO_3^-$ concentration was determined with an ion chromatography system (Dionex-ICS-4000 and Dionex-AS-23/-24, Thermo Fisher Scientific Inc., Waltham, MA, USA). $NH_4^+$ content was

measured with a mass spectrometer (Perkin Elmer SCIEX ELAN 6000, PerkinElmer, Inc., Waltham, MA, USA) until 2016 and afterwards via a continuous flow analysis (CFA, Skalar Analytic GmbH, Erkelenz, Deutschland).

The gaseous $N_2O$ emissions were determined by collecting air samples each week for all lysimeter at Selhausen and for three of the six lysimeters at Rollesbroich—Ro_Y_011, Ro_Y_014, Ro_Y_015—during the period from September 2013 until December 2018. Additional measurements were done within three days after fertilizer application. The samples were manually collected with closed static chamber measurements [39]. Each measurement lasted 32 min, during which four samples of the air in the chamber were taken after 2, 12, 22, and 32 min. The samplings were carried out at around noon. This time of the day was proposed to allow for a good estimation of the mean daily emission rate [44]. The $N_2O$ concentration in the gas samples from the static chambers was analyzed by gas chromatography (Clarus 580 with ECD, PerkinElmer, Inc., Waltham, MA, USA).

The management of the vegetation on the lysimeters—date for the grass cuts, fertilizer application—follows the extensive management of the surrounding grassland in Rollesbroich—i.e., three to four grass cuts per year, low N-fertilizer input by liquid manure. Only for 2018, the date of the organic fertilizer input was missed, and instead of liquid manure, calcium ammonium nitrate with a N-concentration of 27% (CAN27) was added at the end of the year (see Table A1 in the Appendix A). Samples of the mowed grass and applied fertilizer were weighed with a precision scale (EMS 6K0.1 model, KERN & SOHN GmbH, Balingen, Germany, range: 0–6 kg, resolution: 0.1 g) and dried at 60 °C for 24 h and then at 105 °C to evaluate the percentage of dry matter. The dry mater samples were also analyzed with a CHN—carbon hydrogen nitrogen—elemental analyzer (VarioElCube, Elementar Analysensysteme GmbH, Langenselbold, Germany) and a thermal conductivity detector to measure the N-content.

Changes in soil organism dynamics were not monitored as this would involve destructive soil sampling. Plant growth dynamics were monitored via height measurements. A presentation of the gap-filled height data is presented in Figure A2 (see Appendix A).

Climatic variables and data gathered from the lysimeter balances and sensors had previously been processed and checked for plausibility [45,46]. To reduce the noise of the data, the smoothing filter AWAT—Adaptive Window and Adaptive Threshold filter [47,48]—was used. For more information regarding the data preparation, see Pütz et al. [39] and Rahmati et al. [37].

For this study, the data qualified as "good" was downloaded from the TERENO online database—http://teodoor.icg.kfa-juelich.de/ibg3searchportal2/index.jsp, accessed 1 April 2020. When data regarding climatic variables were missing for the reference climatic station of the lysimeters—TERENO-portal, Rollesbroich: Ro_BKY_010; Selhausen: Se_BDK_002—we gap-filled the time-series using data from other stations near the lysimeters from the TERENO-portal and did a step-wise gap-filling: (a) a linear regression was done with each station for each variable; (b) the regression with the highest coefficient of determination ($R^2$) was used first to fill the gaps. The data was processed with R, a programming language and free software environment for statistical computing [49].

### 2.2. Water Balance

The water balance at the water- and energy-limited site was described according to Pütz et al. [39]:

$$Prec - ETa - Dnet = \Delta WS, \tag{1}$$

$$Dnet = Dr - CR, \tag{2}$$

with Prec (mm), ETa (mm), Dnet (mm) is the net water flux across the lysimeter bottom, Dr the seepage water (mm), CR the water injected at the bottom of the lysimeter (mm), mimicking the capillary rise of water, and $\Delta WS$ the change in soil water storage (mm). Prec also includes water from non-rainfall events like dew or fog formation, which can account annual up to 8% of the total Prec at the sites (see Groh et al. [50] and Brunke et al. [51]).

Gaps in the time series' of daily ETa (or Prec) data were filled in a step-wise procedure based on linear regressions with the ETa (or Prec) of the other lysimeters of the same study site, as explained in detail in Groh et al. [40]. For filling all remaining gaps of ETa time series, a linear regression between the data from the lysimeter with those of the grass height-adjusted reference evapotranspiration (ETcrop) was used. ETcrop was calculated on a 10 min basis according to Allen et al. [52] using the full form Penman–Monteith model (see Rahmati et al. [37]).

To fill gaps in the time series' of daily grass height, we applied the following method: (a) we calculated the mean of all the available measurements taken at the first day of the growing period (respectively after the grass cut) and used it as default value for the first day of the growing period (respectively after the grass cut) when no data was available; (b) between two cuts during the growing period, we did a linear interpolation between the available measurements; (c) during the growing period, when there were a data gap between the last grass measurement and the day of the grass cut, we did an extrapolation— based on the last available values. Rahmati et al. [37] did directly an interpolation between the grass height measurements. For this reason, our grass height and ETcrop data differ from theirs.

At the catchment scale, climate–vegetation interactions can be defined according to the Budyko framework [53]. Briefly, plant transpiration and soil evaporation are dependent on the potential evapotranspiration to water supply ratio (i.e., AI, the aridity index, Equation (3)). Indeed, if AI < 1, evapotranspiration is limited by its demand for energy, whereas for AI > 1, water is the limiting factor for evapotranspiration. The other Budyko factor (EI, the annual evaporation index, defined as the ratio of ETa to water supply, Equation (4)) represents the water partitioning between evapotranspiration and run-off In their study, Rahmati et al. [37] observed a mean annual AI < 1 for Rollesbroich—thereafter called E-limited—and AI > 1 for Selhausen—thereafter called W-limited. They, however, defined water supply as the annual precipitation. Thus, their analysis did not account for the effects of ground and surface water interaction, which can lead to strongly biased results [54]. To analyze the climatic conditions at both sites for each year, we therefore used the Budyko framework, as adapted by Condon and Maxwell [54] (Equation (5)), with ETa and ETcrop (mm), and Prec (mm) for the year n. To capture all available water for the soil for the year n, groundwater contribution to water input was added to the denominators: CR (mm) for the year n and $\Delta W$ when the $\Delta WS$ of the year n-1 was positive. The Budyko curve was computed using the equation of Fu [55]. The value of Fu's parameter ($\omega$) was taken fromRahmati et al. [37]—$\omega$ = 2.6.

$$AI_n = \frac{ETcrop_n}{(Prec_n + CR_n + \Delta W)},\tag{3}$$

$$EI_n = \frac{ETa_n}{(Prec_n + CR_n + \Delta W)}\tag{4}$$

$$\Delta W = \begin{cases} \Delta WS_{n-1}, & \Delta WS_{n-1} > 0 \\ 0, & \Delta WS_{n-1} \leq 0 \end{cases}\tag{5}$$

*2.3. Nitrogen Balance*

The soil N-balance for the lysimeter soils was described according to Klammler and Fank [56] by the sum of all measurable input and output fluxes that are balanced by the storage change as

$$N\_WD + N\_Fert - N\_Vol - N\_PU - N\_Dnet - N\_N_2O = \Delta N\_S,\tag{6}$$

$$N\_Dnet = N\_Dr - N\_CR,\tag{7}$$

with N_WD the N from the wet deposition (kg ha$^{-1}$), N_Fert the fertilizer N (kg ha$^{-1}$), and N_Vol the ammonia volatilization of the fertilized N (kg ha$^{-1}$), N_PU (kg ha$^{-1}$); N_Dr is

the leached N in the drainage water (kg ha$^{-1}$) determined from the N-concentration in the bottle where seepage water aliquot is collected, N_CR is the N in the solution injected at the bottom of the lysimeter (kg ha$^{-1}$) determined from the N-concentration in the lysimeter tank, N_N$_2$O is the gaseous flux of N$_2$O (kg ha$^{-1}$), and $\Delta$N_S the change in soil N-storage (kg ha$^{-1}$). For periods without available N$_2$O data, the N$_2$O flux was assumed to be zero because it is mostly negligibly small compared to other N-fluxes in extensively managed grasslands [22,30]. The N-fluxes from N_Dr, N_CR, and N_WD data were processed and gap-filled following the method presented by Knauer [57]. During dry periods, it was not possible to collect enough water to analyze the water samples because the water flow across the lysimeter bottom was small, which led to larger gaps in the concentration time series. There is therefore no observation for 2012 for W-limited. The data time series contain other gaps that correspond to periods with negligible water and nitrogen fluxes.

With the N$_2$O-concentration data—from the gas-chambers of the W-limited lysimeters and the E-limited lysimeters Ro_Y_011, Ro_Y_014, and Ro_Y_015, the N2O flux rate (f0) was computed according to the Hutchinson–Mosier Regression (HMR) approach [58] using the HMR package of R [59]. Either a non-linear HMR or a linear regression was selected according to the criteria described in Deppe [22]. From f0 (ppm min$^{-1}$), the N_N$_2$O flux rate (N_N$_2$O-rate, in µg N m$^{-2}$ h$^{-1}$) was calculated (Equation (8)).

$$\text{N\_N}_2\text{O} - \text{rate} = \text{f0} \times \frac{\text{M\_N}_2\text{O}}{\text{V\_N}_2\text{O}} \times \frac{\text{M\_N} \times 2}{\text{M\_N}_2\text{O}} \times \frac{60}{1000} \qquad (8)$$

with M_N$_2$O the molar mass of N$_2$O (g mol$^{-1}$), V_N$_2$O its molar volume (m$^3$ mol$^{-1}$), M_N the molar mass of N (g mol$^{-1}$), and the numbers are for unit conversions. Daily N_N$_2$O emissions were calculated from the hourly N_N$_2$O emission rates. Then, the N_N$_2$O emissions between two sampling dates—for intervals inferior to one month—were filled using a linear interpolation by averaging the fluxes of two consecutive sampling dates, multiplied by the number of days in this period. With this technique, we obtained the N_N$_2$O emission data for 120 days in 2013, 365 days in 2014, 2015, and 2016, 339 days in 2017, and 142 days in 2018.

To obtain an approximation of N_Vol, the equation of Menzi et al. [60] was used:

$$\text{N\_Vol} = (19.41\text{TAN} + 1.1\text{SD} - 9.51) \times \left( 0.02 \times \left( \text{AR} \times 10^{-3} \right) + 0.36 \right) \qquad (9)$$

with N_Vol in kg N ha$^{-1}$, TAN the total ammonia-N content of the fertilizer (g N ha$^{-1}$), SD the mean saturation deficit of the air (mbar) and AR the application rate of the fertilizer—the amount of each fertilizer application (kg ha$^{-1}$). The environmental variables correspond to the mean value for the day of the fertilizer input. We assumed that TAN made up about 65% of the fertilizer's N [61].

For the CAN27, the approach suggested by the COMIFER Association—the French Committee for the Study and Development of sustainable Fertilization—[62] was used to evaluate the volatilization. Both methods serve in the first instance as a rough estimate rather than a precise method to evaluate N_Vol. We still assumed that this approximation would allow for a more exact calculation of $\Delta$N_S than disregarding N_Vol.

### 2.4. Statistical Analysis

The shapiro.test and leveneTest functions from the stats [49] and car [63] packages in R were used to evaluate the dependent variables' normal-like distribution and the homogeneity of their variance. A $p$-value $< 0.01$ was considered as significant. When necessary, the variables were transformed to obtain a normal distribution using the BoxCoxTrans function from the caret package [64].

The parametric Pearson (respectively non-parametric Kendall) correlation coefficient was used to compare the influence of plant aboveground net primary production (ANPP) and plant N-concentration on N_PU (respectively Dr and N-concentration in Dr on N_Dr). The tests were done with the function cor.test from the package stats.

The interpolation of N_Dr according to Dr was done using a Locally Estimated Scatterplot Smoothing (LOESS) with a span of 0.75 using the function geom_smooth from the package ggplot2 of R [65]. To evaluate the influence of the site factor (W-limited or E-limited) on nitrogen loss, we looked at three elements of the nitrogen balance: N_N$_2$O-rate, N_Dr, and N_PU per harvest. We also took into account the year factor as control variable as well as a random factor specific for each lysimeter. As the data corresponds to observations from the same sites, the assumption of independence necessary for ordinary least square regression was not met. For this reason, the lmer function from the lmerTest package [66] was used with the anova function from the car package (Analysis of Variance Table of type III with Satterthwaite's method) to perform a mixed-effects analysis for repeated-measures.

When analyzing N_PU, the harvest number of the year was also used as a control variable.

When analyzing N_Dr, Dr was also taken into account as control variable. Even once transformed, the N_Dr data did not meet the assumption of normality and homoscedasticity of variance according to the Shapiro and Levene tests. However, the sample size was very important—e.g., 1498 observations for E-limited in 2014. This affected negatively the *p*-value calculation of those two tests. We therefore still implemented the parametric mixed-effects analysis, bearing in mind that the results may not be as trustworthy. Indeed, according to the quantile-quantile plot (Figure A3) the data departed in some subgroups from a normal-like distribution.

## 3. Results

### 3.1. Water Balance and Effects of Different Climatic Conditions

The mean annual Prec rates for the period 2012–2018 were 1060 mm year$^{-1}$ for E-limited and 665 mm year$^{-1}$ for W-limited (Tables A2 and A3, see Appendix A). The Prec made up 97% of the water input for E-limited and 88% for W-limited (Figure 1A, Figure 2A, and Figure 3A). Mean annual Prec was always higher for E-limited than for W-limited. The mean monthly Prec was always higher for E-limited, especially during the winter months (Figures 1A and 3A) and ranged from being 51% higher than that for W-limited in January to 22% higher in August. The inter-annual coefficient of variation of Prec was higher at the water-limited site: 12% for W-limited and 8% for E-limited. The coefficient of variations of the mean monthly Prec (Figure 1) was also slightly higher in Selhausen: 23% for W-limited compared to 20% for E-limited.

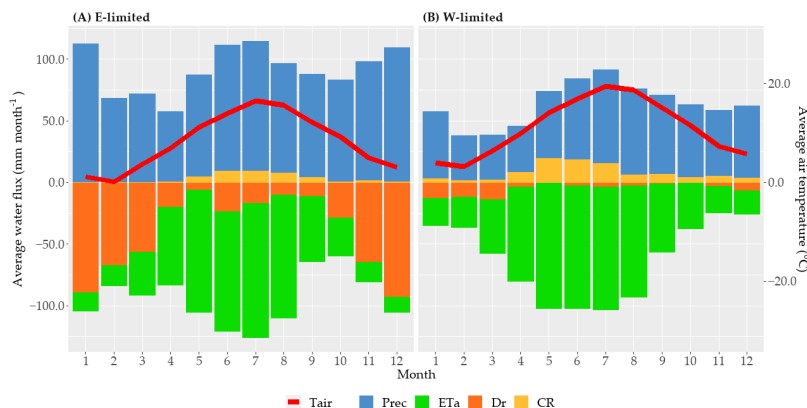

**Figure 1.** Mean monthly values of the precipitation (Prec), actual evapotranspiration (ETa), drainage (Dr), upward water flow (CR), and average air temperature (T$_{air}$) for (**A**) Rollesbroich the energy-limited site (E-limited) and (**B**) Selhausen the water-limited site (W-limited). Positive values represent an input and negative values a loss of water for the system.

Most of the lysimeter drainage—76% of total Dr for E-limited and 80% for W-limited—occurred between beginning of November and end of March (Figures 1 and 3B). This period corresponds to the non-growing period when the soil water demand of the grass

was relatively low. The fraction of Dr made up 44% of the gross water input for E-limited and 8% for W-limited. The CR flux was also higher for W-limited as compared to E-limited (Figures 2C and 3B). The mean annual rate of ETa was similar for both sites—650 mm year$^{-1}$ for E-limited and 700 mm year$^{-1}$ for W-limited (Figure 1A,B, Figure 2D, and Figure 3A). This explains why the yearly ΔWS was usually lower for E-limited in spite of the wetter climate (Figure 2F). The year 2014 was wetter than the average of the observation period and the soil WS increased in lysimeters at both sites (Figure 2E) leading to relatively high soil water contents. The year 2018 was drier than the average of the observation period as indicated by the lowest ΔWS value of the studied years.

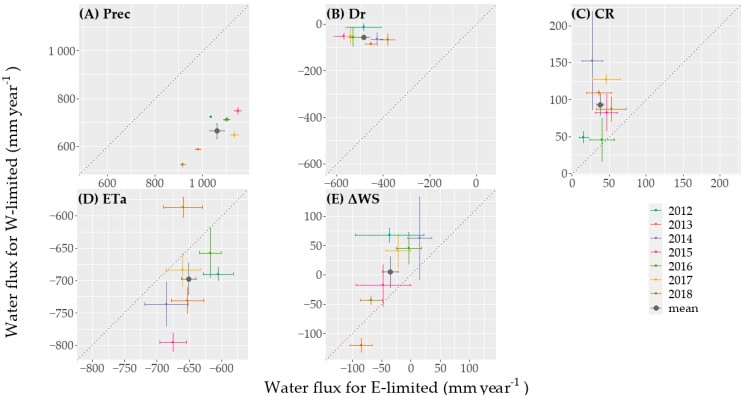

**Figure 2.** Comparison of the annual water balance components for lysimeters with soils from Rolles-broich located in the energy-limited site Rollesbroich (E-limited, x-axis) and the water-limited site Selhausen (W-limited, y-axis) for (**A**) precipitation (Prec), (**B**) drainage (Dr), (**C**) upward directed water mimicking capillary rise (CR), (**D**) actual evapotranspiration (ETa), and (**E**) the resulting soil water storage (ΔWS). Positive values represent an input and negative values a loss of water for the soil lysimeter system. "mean" in the legend indicates the mean annual values for the period 2012–2018 and the inter-annual variability is represented by the standard error of annual means.

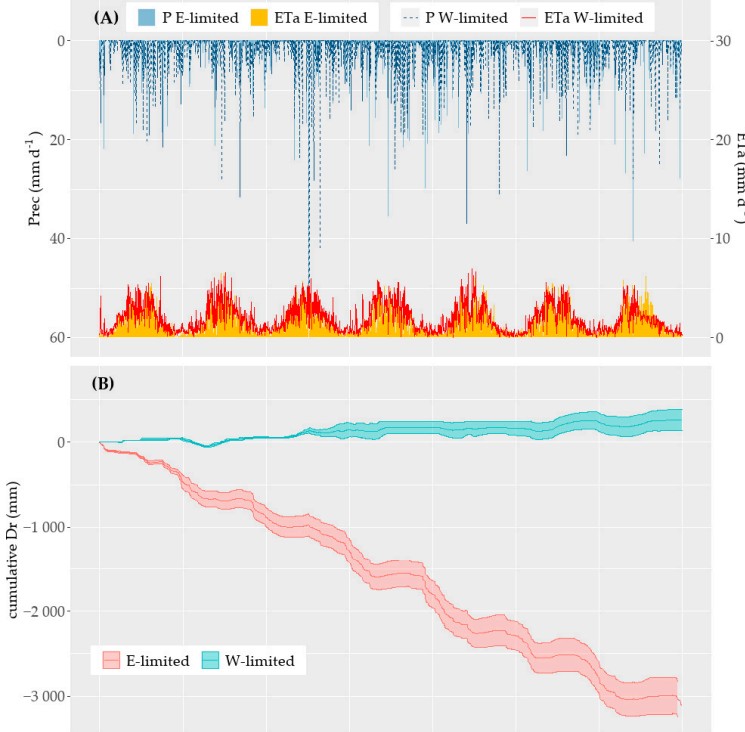

**Figure 3.** *Cont.*

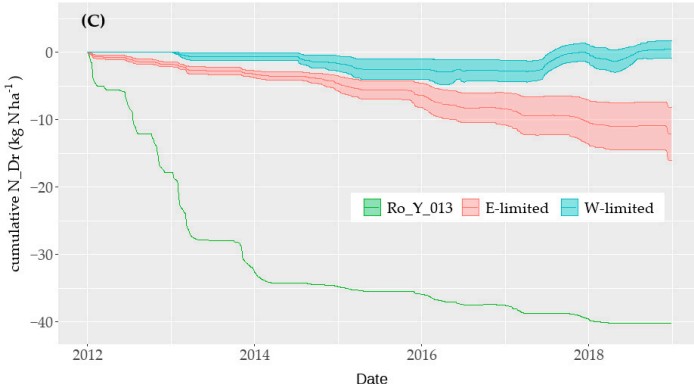

**Figure 3.** For the period between 2012 and 2018 in Selhausen, the water-limited site (W-limited) and Rollesbroich, the energy-limited site (E-limited), (**A**) daily precipitation (Prec) and actual evapotranspiration (ETa), (**B**) cumulative net drainage (Dr), (**C**) cumulative net leached nitrogen (N_Dr). For (B) and (C), the colored areas represent the standard deviation to the mean value of each study site. For (C), because E-limited Ro_Y_013 behaved differently compared with the other E-limited lysimeters, its dataset is presented separately.

For the Budyko plots, the average values of AI and EI were 0.65 and 0.59 for E-limited and 1.04 and 0.88 for W-limited, respectively. According to the Budyko framework, water was more limiting for W-limited than for E-limited. On both sites, the year 2018 had a much higher AI value than the mean value for the observation period: 0.83 for E-limited, 1.40 for W-limited (Figure 4). This led to an increase of the EI for E-limited in 2018. For W-limited, the EI of 0.90 for 2018 was lower than the EI of 0.96 for 2013. This shows that the high AI led to a decrease of ETa for W-limited in 2018. For 2013 for E-limited, ETa > ETcrop, which indicates an underestimation of ETcrop (Table A2).

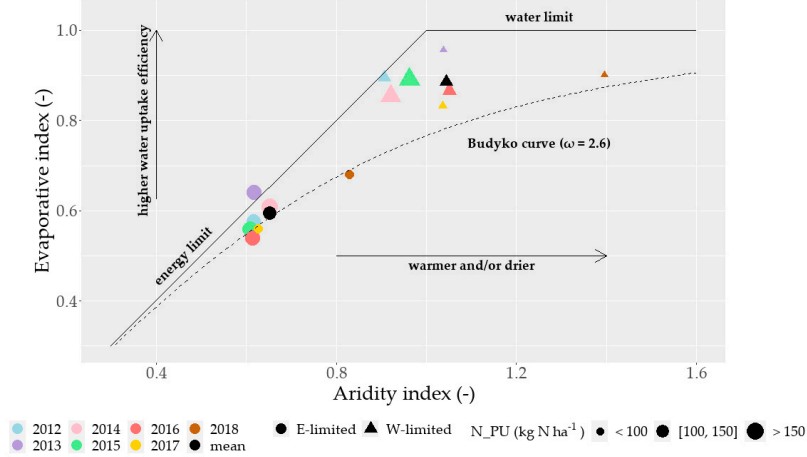

**Figure 4.** Budyko plot for Rollesbroich lysimeters in Rollesbroich, the energy-limited site (E-limited, circles) and in Selhausen, the water-limited site (W-limited, triangles) for the period 2012–2018. The aridity index (x-axis) is the ratio of the cumulated yearly grass height-adjusted reference evapotranspiration to the available water, defined as precipitation and ground water contribution, following the recommendation of Condon and Maxwell [53]. The evaporative index (y-axis) is the ratio of the cumulated yearly actual evapotranspiration to the water input. The points' size correspond to the annual plant nitrogen uptake (N_PU). "mean" in the legend indicates the mean annual values for the period 2012–2018. The dotted curve shows an exemplary Budyko curve, with Fu's parameter [55] taken from Rahmati et al. [37]—ω = 2.6. The diagonal black line is the energy limit for the actual evapotranspiration—actual evapotranspiration is equal to potential evapotranspiration. The horizontal black line is the water limit for the actual evapotranspiration—actual evapotranspiration is equal to the available water.

### 3.2. Nitrogen Balance and Effects of Different Climatic Conditions

The mean annual ΔN_S for the period 2012–2018 ranged between −129 kg N ha$^{-1}$ for E-limited and −98 kg N ha$^{-1}$ for W-limited (Figure 5, Tables A3 and A4). The grassland ecosystem had therefore depleting N-pools at both sites. ΔN_S was 58% lower for E-limited and 82% lower for W-limited in 2014 compared with 2018. The mean annual ΔN_S was thus higher at the drier site and during drier years. Moreover, the yearly N-loss decreased during the study period. For the period 2012–2018, the mean annual N_PU made up about 86% of the N-loss for E-limited and W-limited. The mean annual N_PU for the period 2012–2018 ranged between 161 kg N ha$^{-1}$ for E-limited and 130 kg N ha$^{-1}$ W-limited. The mean annual N_PU was higher at both sites in 2014—a wet year—compared to 2018—a very dry year: 40% change from wet to drier year for E-limited and 54% for W-limited (Figures 4 and 5). Generally, especially for W-limited, higher AI was linked with a lower N_PU (Figure 4). The Pearson correlation coefficient between N_PU and ANPP for all the lysimeters equaled 0.82, while it was 0.34 for the correlation between N_PU and N-concentration in the plant. The increase in N_PU was therefore mainly linked with an increase in ANPP rather than an increase in N-concentration in the plant.

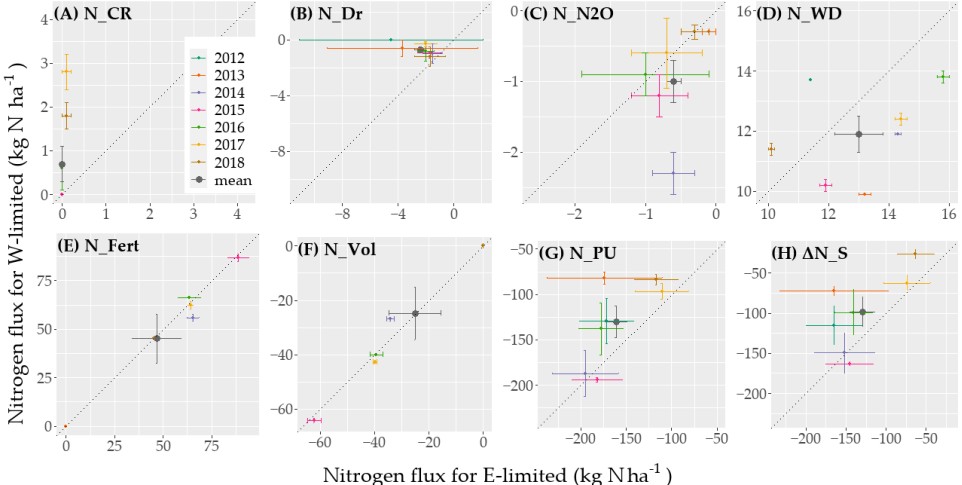

**Figure 5.** The different components of annual nitrogen balance for lysimeters from Rollesbroich in Rollesbroich the energy-limited site (E-limited, x-axis) and Selhausen the water-limited site (W-limited, y-axis) for (**A**) nitrogen in the upward directed water (N_CR), (**B**) nitrogen in drainage water (N_Dr), (**C**) nitrogen lost by $N_2O$ emissions (N_N2O), (**D**) nitrogen in wet deposition (N_WD), (**E**) nitrogen input by fertilizer (N_Fert), (**F**) ammonia volatilization of the fertilizer nitrogen (N_Vol), (**G**) plant nitrogen uptake (N_PU), and (**H**) the resulting soil nitrogen storage (ΔN_S). The negative values represent a nitrogen loss for the lysimeters; "mean" in the legend indicates the mean annual values for the period 2012–2018 and the inter-annual variability is represented by the standard error of annual means. There are gaps in the $N_2O$ emission data for the years 2013 and 2018 resulting in lower yearly emissions.

For the period with fertilizer application—2014–2018, mean annual N_Fert made up to 83% of the N-input for E-limited and 82% for W-limited.

The mean N_PU per harvest was higher for the first cut of the year than for the other cuts, with 64 kg N ha$^{-1}$ for E-limited and 67 kg N ha$^{-1}$ W-limited (Table A6). For the first cut, the N_PU was slightly higher for W-limited, despite a lower N_PU rate, because of the earlier start of the growing period in Selhausen. For the three other cuts, the mean N_PU was higher for E-limited because of the water limiting conditions for W-limited.

According to the mixed-effect analysis, the location factor (W-limited or E-limited) was found to have a moderately significant effect on N_PU ($p$-value < 0.05), while the interaction between (a) the location and the harvest number and (b) the location and the year had both a significant effect ($p$-value < 0.01).

During the years for which there were no gaps in the $N_2O$ data—2014–2017, mean annual $N\_N_2O$ made up about 0.4% of the N-loss for E-limited and 0.6% for W-limited. The mean $N\_N_2O$-rate between September 2013 and December 2018 was 8.6 μg N m$^{-2}$ hour$^{-1}$ for E-limited and 13.9 μg N m$^{-2}$ hour$^{-1}$ for W-limited. After the peak of $N\_N_2O$ emission for W-limited in 2014, there was a decrease in emission over the years from 2.3 kg N ha$^{-1}$ in 2014 to 0.6 kg N ha$^{-1}$ in 2017. The lysimeter E-limited Ro_Y_015 emitted in total 14.3 kg N ha$^{-1}$ of gaseous $N\_N_2O$ between September 2013 and December 2018, which were significantly higher than the mean $N\_N_2O$ of 5.7 ± 0.58 kg N ha$^{-1}$ of the other two E-limited lysimeters. This led to a high standard deviation between the lysimeters for this group.

According to the mixed-effect analysis, the location factor was not found to have a significant effect ($p$-value > 0.05) on $N\_N_2O$-rate as opposed to the interaction between the year and location factors which had a significant effect ($p$-value < 0.01).

The Kendall correlation coefficient between N_Dr and Dr for all the lysimeters was equal to 0.72 ($p$-value < 0.01), and to 0.37 ($p$-value < 0.01) for N_Dr and the N-concentration in the leached water. Higher N_Dr for E-limited as compared to W-limited was thus caused by a higher Dr rather than a higher N-concentration (Figures 3C and 6). Indeed, for W-limited, because Dnet was almost zero, N_Dnet remained also around zero (Figures 3 and 4; Tables A3 and A5). As a result of extensive management and the high N-uptake and in spite of the high Dr for E-limited—482 mm year$^{-1}$, N_Dr was low as well: 2.4 kg N ha$^{-1}$ year$^{-1}$ (Tables A2 and A4).

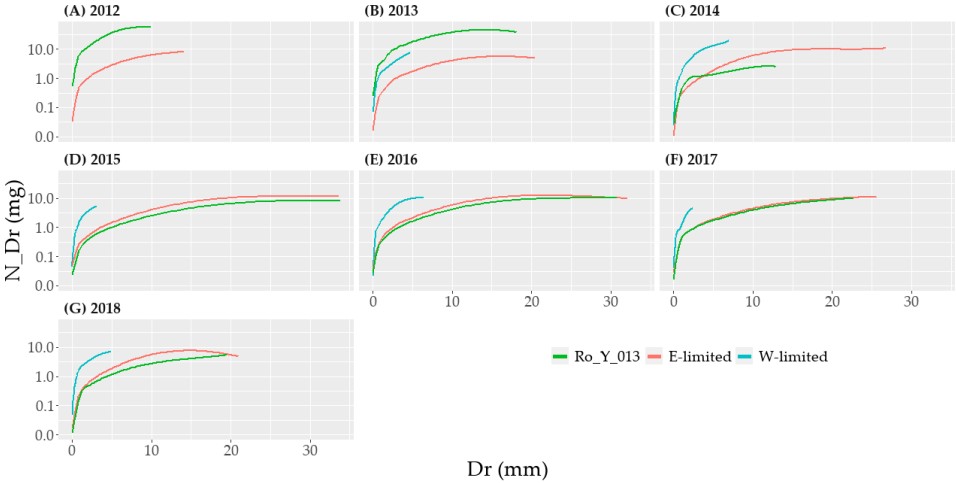

**Figure 6.** Interpolated relations of nitrogen mass in drainage (N_Dr) versus the amount of drainage (Dr) from the Rollesbroich lysimeters obtained from data pairs of the 14-days sampling periods between 2012 and 2018, for the lysimeters in Selhausen, the water-limited site (W-limited), the lysimeter Ro_Y_013 and the other lysimeters in Rollesbroich, the energy-limited site (E-limited). Because E-limited Ro_Y_013 behaved differently compared with the other E-limited lysimeters, its dataset is presented separately. The interpolation was done using a Locally Estimated Scatterplot Smoothing function with a span of 0.75. No observations of N_Dr were available for Selhausen in 2012.

The input of organic N_Fert did not lead to a significant increase in yearly N_Dr. However, for E-limited, the second highest daily N_Dr occurred in December 2018, following the only input of mineral N_Fert. For W-limited, Dr started later in the season than for E-limited, so the leaching of the mineral N_Fert might have occurred in beginning of 2019.

In 2012 and 2013, the high concentration of nitrogen in Dr for E-limited Ro_Y_013 (Figures 3C and 6) led to a higher average N_Dnet compared to other E-limited lysimeters (Figure 7). Because of this, there was a high standard deviation of the mean N-concentration for E-limited (Figure 8) in Dr. For E-limited Ro_Y_015, we observed a period of larger increase of N_Dnet between values of 900 mm to 1100 mm and around 3000 mm (Figure 7). Otherwise, this lysimeter did not show a different N_Dnet dynamic. Finally, we observe a high average N-concentration in Dr for 2015 compared with the other years (Figure 8).

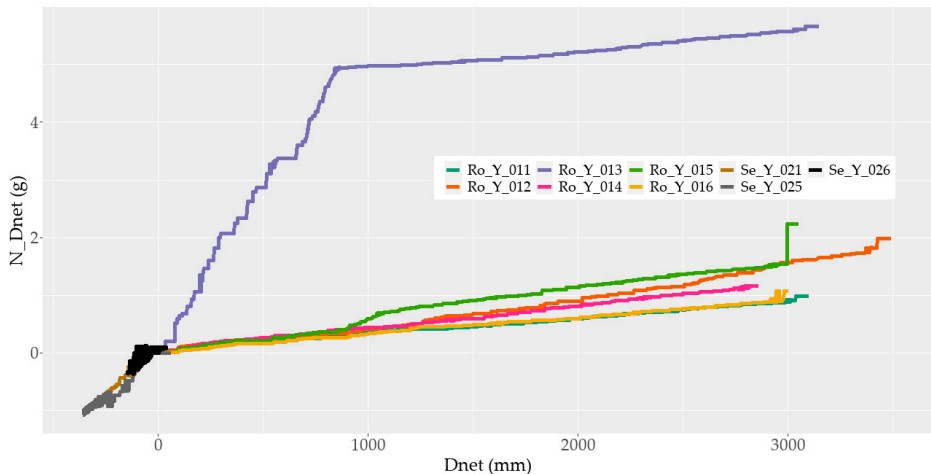

**Figure 7.** Cumulated net drained nitrogen (N_Dnet) versus cumulated net drainage (Dnet) in the Rollesbroich soil for the lysimeters in Selhausen, the water-limited site (W-limited) and the lysimeters in Rollesbroich, the energy-limited site (E-limited). Negative values correspond to periods when water and nitrogen was entering the system.

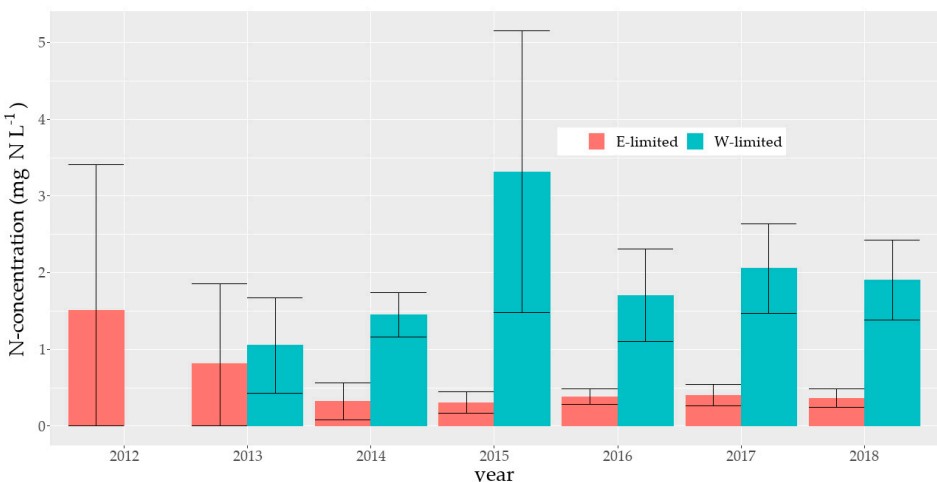

**Figure 8.** Mean and confidence interval (95% confidence level) of nitrogen concentration (mg N $L^{-1}$) in the drained water per year for the Rollesbroich soil in Rollesbroich, the energy-limited site (E-limited) and Selhausen, the water-limited site (W-limited). No observations of N-concentration were available for Selhausen in 2012. The confidence intervals per study site were calculated from the lysimeters' annual means.

The N-concentration in Dr was not significantly higher for W-limited compared to E-limited in 2013. For 2014 onwards however, the confidence intervals—95% confidence level—of N-concentration in W-limited and E-limited do not overlap (Figure 8).

According to the mixed-effect analysis, even when leaving aside the results from the lysimeter Ro_Y_013, the influence of the location factor was not significant ($p$-value > 0.05). However, the interaction factors between (a) the location and the year and (b) the location and N_Dr had a significant effect.

## 4. Discussion

### 4.1. Space-for-Time Substitution Approach

Due to the different gap-filling procedure compared to Rahmati et al. [37] for the grass height data that was used to estimate ETcrop, the annual ETcrop was on average 1% lower at E-limited and 3% lower at W-limited compared to their study. Differences were higher during the non-growing period. Further differences between the AI and EI values found by

Rahmati et al. [37] and those in this study result from the different calculation method for the denominator of the indices. Our results on the Budyko plots demonstrate that the grassland ecosystem was shifted from a more demand—energy (potential evapotranspiration)—limited to a rather supply—water—limited system. This was also reported by Rahmati et al. [37], using however the classical Budyko framework—without taking into account groundwater–surface water interactions. The Budyko-framework suggests that the grassland ecosystem W-limited is in transition from a demand- to and supply-limited system, which in turn have an impact on other important ecosystem functions such as ANPP, groundwater recharge or water quality [41].

Comparing the E- and W-limited sites as a SFT substitution approach allowed us to evaluate the effects of a climatic shift foreseeable in the northwest of Germany in the coming years. As responses of the N-cycle vary according to the intensity of the change [29], those results cannot be extended to stronger variations. Moreover, comparing E-limited and W-limited does not allow us to take into account the effects of climate change related to an increase in atmospheric $CO_2$ concentrations, which should affect the water balance, ANPP, and net N-mineralization rate [12,41]. Further, the translocation-induced climatic change was abrupt and strong in its expression, in contrast to long-term environmental variations which are, excluding extreme years, more gradual [40], which probably affects the dynamics of change in the N-cycle differently. Moreover, the larger hydrogeological setting [67] and the depth to an impermeable layer differ between both sites [68,69]. As discussed by Groh et al. [68], this could cause a strong bias in water fluxes across the tension-controlled lysimeter boundary, and therefore affect the N-cycle.

Climate change already has a noticeable effect on the processes of both sites. Indeed, as found by Rahmati et al. [37], there has been, since 2015, a decrease in the minimum soil water content of both sites. The year 2018 was also an especially warm, sunny, and rain-deficient year [37,38]. Moreover, we could observe that both sites suffer from a net loss of N. This shows that the current status corresponds to a transitory period. As explained by Thornley and Cannell [11], under constantly changing climatic conditions, N-poor grasslands go through a continual adjustment toward a never-reached steady state. Therefore, the long-term response of the studied sites with respect to N-loss and mineralization rate could differ in magnitude and sign to the ones observed during the study period.

Finally, until 2013, N-concentration in Dr for E-limited Ro_Y_013 still showed the effects of the environmental drivers dating from before 2011—start of the monitoring period (Figures 3C and 6). It may be relevant to only take into account the data gathered after 2013 for the analysis of the translocation on the N-cycle dynamic.

*4.2. Net Mineralisation Rate*

The mixed-effect analysis showed that the location factor (W- or E-limited) had a significant influence on N_Dr for a set level of Dr. As seen in Figure 6, the N-concentration at W-limited is indeed higher for a particular Dr. This may seem surprising, as lower soil water content is usually found to decrease net mineralization [23]. However, Meisser et al. [26] postulated that an increase in soil inorganic N occurred when N_PU decreased more than the gross N-mineralization rate under a drier and warmer climate. This is in accordance with the findings of this study, as N_PU was lower for W-limited. Ineson et al. [19] observed a decrease in net mineralization rate and an increase in N_PU under drier and warmer conditions, while Fu et al. [14] observed an increase in net mineralization and a decrease in N_PU. This seems to confirm that the effect of climate change on N_PU is the main driver of the variation in net mineralization rate.

As shown by Rahmati et al. [37], there is a stronger annual variation in the soil water content for W-limited than for E-limited. This might also result in higher soil mineral N-concentration for W-limited (Figures 7 and 8). Indeed, as explained by Borken and Matzner [29] in their meta-analysis, drying-wetting cycles may cause a higher cumulative N-mineralization rate in temperate grasslands.

After the wettest period of the study (end of 2014 to beginning of 2015) we observed the highest N-concentration in leached water for W-limited. Although N_Fert input might have been one of the causes—see Section 4.3 of the discussion, a complementary explanation is given by Risch et al. [33] in their meta-analysis of nitrogen mineralization in grasslands: they observed that $T_{air}$ during the wettest quarter of the year had a strong positive influence on the net mineralization rate. Indeed, $T_{air}$ increases microbial activity when soil humidity is also elevated, which was the case in 2014. Therefore, it seems that the higher $T_{air}$ linked with the inter- and intra-annual variability of Prec caused the higher net mineralization, because of mineralization pulse and decreased N_PU. This confirms our first hypothesis that the net mineralization rate is higher under warmer and drier conditions.

*4.3. Nitrogen Leaching*

The high N-concentration in Dr for Ro_Y_013 might have been caused by the history at E-limited—e.g., more fertilizer applied for Ro_Y_013 than at the other lysimeters. The Dr for W-limited Ro_Y_013 in 2013 may thus be from older water percolating in the soil monoliths before 2010. The N-concentration of Dr represented that of the soil solution from before the translocation. This would explain why the N_Dr concentration in N_D are not significantly different between the two sites in 2013 according to the 95% confidence interval (Figure 8).

For E-limited Ro_Y_015, the transitory quick increases in N_Dnet could indicate the existence of preferential flow for this lysimeter. Finally, the wet year 2014 for W-limited, and the two N_Fert application during the non-growing period (Table A6), might have caused an increase in N-concentration occurred at this site, leading to a higher N-concentration in Dr for 2015 compared with the other years (Figure 8).

The mean annual N_Dr of 2.4 kg N ha$^{-1}$ year$^{-1}$ for E-limited is within the range of observed N_Dr by Fu et al. [14] for extensively managed grassland—2.04–6.91 kg N ha$^{-1}$ year$^{-1}$. However, under a warmer and drier climate the average value for N_Dr for W-limited was with 0.7 kg N ha$^{-1}$ year$^{-1}$ clearly below this range. Like Fu et al. [14], we observed a decrease of Dr at the drier and warmer site. They, however, observed no clear effects of a drier and warmer climate on N_Dr in case of extensive management. This is caused by the fact that the Dr reduction following translocation was not as important as it has been for our study at the water-limited site.

In Fu et al. [14], drier and warmer climatic conditions led to an increase of N_Dr in case of higher N_Fert. In 2018, the only year with mineral fertilizer input, the rewetting of the soil in Selhausen occurred in December. The leaching of N_Fert likely occurred after the period of the study. We could therefore not see if the effect of the climatic gradient on the mineral fertilizer leaching was similar to the effects observed by Fu et al. [14] for intensively managed grasslands.

In addition, in our study increased N-leaching was caused by the higher downward water flow rather than by higher N-concentration. In accordance with other observations, risks of nitrate leaching during the growing season were lower, in spite of the manure addition during this period [14,20]. This was explained by the higher ETa and the corresponding uptake of N by plants. Longer growing periods for W-limited with a larger demand for ET and lower Prec slowed down the displacement of solutes, increased N-concentration in the upper soil layer and led thus to a decrease in N-leaching.

In consequence and contrary to our hypothesis, low organic N_Fert input in combination with the higher mineralization rate at the warmer and drier site did not lead to a faster depletion of the mineral nitrogen pool by leaching.

However, in spite of the low N_Fert, we still observed a higher N-concentration in Dr for W-limited compared to E-limited. The year with the highest N-concentration occurred in 2015 (Figure 8) after the wettest year of the study (Figure 2) and following two N_Fert inputs during the non-growing period (Table A1). Therefore, as stated by Fu et al. [14], it is essential to apply N_Fert early in the growing period of the grass to avoid groundwater N-pollution.



### 4.4. N$_2$O Emissions

N_N$_2$O was negligibly small as compared to the other N-fluxes, as it made up less than 1% of the total annual N-balance component "N loss" (Figures 2 and 5). Therefore, for the years 2012, 2013, and 2018, the underestimation of the N_N$_2$O flux did not lead to a significant underestimation of the yearly N-losses.

As observed by Deppe [22], soil N_N$_2$O emissions have a high spatial variability, leading to large standard deviation in the mean yearly N_N$_2$O emission flux. A larger number of lysimeters with static chambers would therefore be necessary in order to detect smaller differences between the two study sites. Following the observations of Huang et al. [70], we observed a higher average N_N$_2$O emissions for one E-limited lysimeter with preferential flow. This may have been caused by increased denitrification along flow paths [24], and this increased the mean and standard deviation of E-limited N_N$_2$O emissions. Consequently, it was difficult to detect small variations in N_N$_2$O emissions between E-limited and W-limited.

Soil N_N$_2$O emission in grasslands also have a high temporal heterogeneity [16,22], and weekly measurements may not allow the detection of short emission events. Deppe [22] found that, for crops cultivated on sandy loam, 40% of the annual emissions were linked to 5–10% of the measurements. Increasing the numbers of measurements after drying-wetting cycles could help detect more of such potential high emission events.

Contrary to the findings of Bell et al. [30] but in accordance with the observations of Cardenas et al. [21], N_N$_2$O emissions following fertilizer input did not seem to have a significant influence on yearly N_N$_2$O emissions in temperate grasslands receiving less than 100 kg N ha$^{-1}$ year$^{-1}$ N_Fert. However, as observed in their studies, N_N$_2$O emission peaks may occur several weeks after fertilizer input. Even though N$_2$O emissions were measured within three days after N_Fert input, it is therefore possible that other fertilizer-induced emission events were missed. The observed range of total yearly N_N$_2$O emissions in our study of 0.1–2.3 kg N ha$^{-1}$ year$^{-1}$ is similar to the 0.2–2.7 kg N ha$^{-1}$ year$^{-1}$ emissions found by Cardenas et al. [21] and Bell et al. [30] for grasslands receiving less than 100 kg N ha$^{-1}$ year$^{-1}$ N_Fert. For E-limited, the emission factor with regard to the fertilizer application was of 1.1%, which is slightly superior to the default emission factor of 1% given by the IPCC [71] for mineral and organic fertilizer. However, the emission factor for W-limited was higher—i.e., 2%.

Although the emissions remained limited in E-limited, the location factor had a significant effect on N_N$_2$O-rate for a specific year. Moreover, in 2014, the difference in yearly N_N$_2$O emission was especially higher for W-limited in comparison to E-limited. During this year, a high mean T$_{air}$, ΔWS and N-concentration in the soil water were observed. This is in accordance with the findings of Guntiñas et al. [35], who observed that the positive effect of soil moisture and temperature was higher when the other variable was elevated as well.

### 4.5. Plant Nitrogen Uptake

The observed yearly N_UP of 130 kg N m$^{-2}$ year$^{-1}$ for W-limited and 161 kg N m$^{-2}$ year$^{-1}$ for E-limited was within the range of the values found in the literature for extensively managed temperate grasslands (127 to 273 kg N m$^{-2}$ year$^{-1}$) [14,34].

In accordance with our initial hypothesis, the average yearly N_PU clearly decreased by 19% when the soil ecosystem from Rollesbroich was exposed to a drier and warmer climate at Selhausen. The high influence of the study site on N_PU is underlined by the mixed-effect analysis, were the location factor alone had a moderately significant effect on N_PU and a significant effect for a specific harvest number. We also observed a change of the intra-annual variability of N_PU after the translocation to a drier and warmer site. Zhang et al. [18] showed for a grassland in China that an increase in temperature led to a decrease in ANPP, and thus to a lower N_PU during summer. The temperature-induced earlier start of the growing period allows a higher ANPP in spring, which follows the observation of Zhang et al. [18] that higher T$_{air}$ at the beginning of the year has a positive

effect on ANPP. With rising $T_{air}$, we should therefore observe an increase in the intra-annual variation of N_PU, with a higher N_PU in spring, when water is non-limiting in this period, compared to a lower N_PU in summer because of water-limited conditions and heat stress. Moreover, the N_PU should start earlier in the year, because of the temperature-induced earlier start of the growing period. This will further increase N_PU in spring and further diminish the soil resources before the start of the summer, leading to a decrease in N_PU in summer.

Zistl-Schlingmann et al. [34] and Fu et al. [14] did not observe a negative effect of warmer and drier conditions on the ANPP and N_PU of grasslands. However, as stated by Zistl-Schlingmann et al. [34], this is likely partly caused by the combination of high soil moisture and mineral N, in contrast to our study sites. The decrease in N_PU at the drier site and during drier years follows the findings of studies which stated that for N-poor ecosystems, when water is limiting, ANPP and the plants' capacity to cover their N-demand decreased [25,27]. This could be caused by a lower gross mineralization and nitrification rate and by a lower nutrient diffusion in the soil [28]. Moreover, Dijkstra et al. [25] found that microbes stay active at lower soil water content than plants. The lower N_PU may thus be caused by the competition with microbes for N-uptake. The occurrence of annual N_PU < 100 kg ha$^{-1}$ in 2013, 2017, and 2018 in Selhausen was also likely caused by the limited Prec: as found by Hatier et al. [72], a minimum of 700 mm year$^{-1}$ is necessary for the growth of *L. perenne*. The temporal occurrence of water stress und low Prec led to an increase in the intra-annual variability of the ANPP and N_PU (Table A6). Indeed, ETa—and thus ANPP and N_PU—undergoes a higher variation at the water-limited site [37].

This variation in the N_PU dynamics of the plant should be taken into account for the management of the fertilizer input, to avoid groundwater N-pollution—see Section 4.3 of the discussion.

## 5. Conclusions

This study compared seven years of high-precision lysimeter data obtained from grassland soil monoliths that were exposed to two different climatic conditions. Three out of nine studied lysimeters were transferred from an energy-limited to a water-limited site, enabling us to study the impact of a changing climatic regime on the soil water and soil nitrogen balance. We tested the hypothesis that a lower evapotranspiration and plant nitrogen uptake combined with periodical droughts will increase the net mineralization resulting in higher soil nitrogen concentrations and increased nitrate leaching or gaseous nitrogen emissions.

Our analysis showed that the transfer from an energy- to a water-limited site affected both the soil water and soil nitrogen balance. The space-for-time substitution approach combined with a detailed monitoring of states and fluxes of water and nitrogen in soil is therefore a useful tool for determining the effects of climate change under natural field conditions.

This change led to a decrease in overall plant nitrogen uptake at the water-limited site despite a relatively stronger increase in the uptake at the beginning of the year. This can have a strong influence on fertilizer use efficiency of the vegetation; therefore, fertilizer application strategies need to be adapted accordingly.

Due to lower plant nitrogen uptake, nitrogen concentration in drainage water for a given drainage volume tended to be higher under drier and warmer climatic conditions than under wetter and colder climatic conditions, which may indicate a higher net mineralization rate.

Contrary to our initial assumption, and although the translocation affected the drainage water nitrogen concentration and the nitrous oxide emission rate, the amount of nitrogen lost through drainage or gaseous nitrous oxide emission was found to be small and almost identical for both energy- and water-limited climates, even for wetter years. This would mean that loss of nitrogen to the atmosphere from gaseous $N_2O$ emissions or/and from nitrate leaching to the groundwater in extensively managed temperate grasslands

remains limited even in the case of changing climatic conditions from an energy-limited to a moderately water-limited ecosystem.

Longer observation periods are needed to monitor the development of the soil nitrogen storage to follow the system's adaption to changing climatic conditions, as a new quasi-equilibrium between organic matter input and mineralization is reached only after many years.

**Author Contributions:** Conceptualization, T.P., J.G. and H.H.G.; methodology, J.G., T.P., M.G., N.B., and H.H.G.; validation, J.G. and T.P.; formal analysis, M.G.; investigation, M.G. and J.G.; resources, J.G. and T.P.; writing—original draft preparation, M.G.; writing—review and editing, M.G., J.G., H.H.G., N.B., H.V., and T.P.; visualization, M.G.; supervision, T.P. and J.G.; project administration, T.P. and J.G.; funding acquisition, T.P. All authors have read and agreed to the published version of the manuscript.

**Funding:** This research received funding from the Helmholtz Association (HGF), the Federal Ministry of Education and Research (BMBF); the German Federal Ministry of Food and Agriculture (BMEL) and the Ministry for Science, Research and Culture of the State of Brandenburg (MWFK).

**Institutional Review Board Statement:** Not applicable.

**Informed Consent Statement:** Not applicable.

**Data Availability Statement:** The data analyzed in this study are openly available on the TERENO-portal at www.tereno.net/ddp, accessed on 1 April 2020.

**Acknowledgments:** We acknowledge the support of TERENO and TERENO-SOILCan. We thank the technical staff of the Forschungszentrum Jülich GmbH, especially Ferdinand Engels, Rainer Harms, Martina Krause, Werner Küpper, Philip Meulendick, Steffi Stork, Leander Fürst, for their excellent, prudent, and persevering work.

**Conflicts of Interest:** The authors declare no conflict of interest.

**Abbreviation**

| | |
|---|---|
| AI | Aridity index |
| ANPP | Aboveground net primary production |
| AR | Application rate of the fertilizer |
| CR | Water injected at the bottom of the lysimeter mimicking the capillary rise of water |
| DIN | Dissolved inorganic nitrogen |
| Dnet | Net seepage water |
| Dr | Seepage water |
| EI | Evaporation index |
| E-limited | Rollesbroich, the energy-limited site |
| ETa | Actual evapotranspiration |
| ETcrop | Grass height-adjusted reference evapotranspiration |
| CAN27 | Calcium ammonium nitrate fertilizer 27% |
| M_N | Nitrogen molar mass |
| M_N$_2$O | Nitrous oxide molar mass |
| N | Nitrogen |
| N_CR | Nitrogen in the solution injected at the bottom of the lysimeters |
| N_Dnet | Net leached nitrogen |
| N_Dr | Leached nitrogen |
| N_Fert | Fertilizer nitrogen input |
| N_N$_2$O | Nitrous oxide nitrogen gaseous emissions |
| N_N$_2$O-rate | Nitrous oxide nitrogen gaseous emissions flux rate |
| N_PU | Plant nitrogen uptake |

| N_Vol | Ammonia volatilization of the fertilized nitrogen |
|---|---|
| N_WD | Wet deposition of nitrogen |
| $N_2O$ | Nitrous oxide |
| $NH_3$ | Ammonia |
| $NO_3^-$ | Nitrate |
| Prec | Precipitation |
| Ro_Y_0X | Lysimeter number X with Rollesbroich soil |
| SD | Saturation deficit of the air |
| Se_Y_0X | Lysimeter number X with Rollesbroich soil translocated to Selhausen |
| SFT | Space for time substitution approach |
| $T_{air}$ | Air temperature |
| TAN | Total ammonia nitrogen content of the added fertilizer |
| V_N$_2$O | Nitrous oxide molar volume |
| W-limited | Selhausen, the water-limited site |
| $\Delta$N_S | Change in soil nitrogen storage |
| $\Delta$WS | Change in soil water storage |

## Appendix A

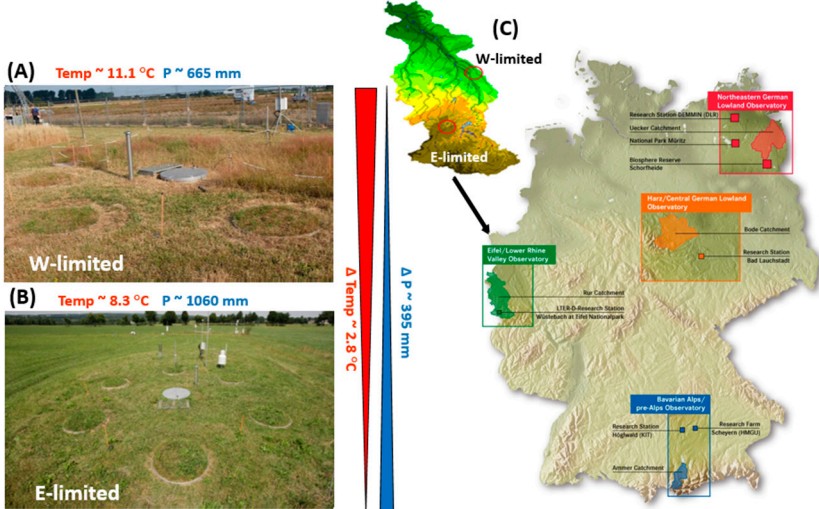

**Figure A1.** Study sites and mean temperature and precipitation of (A) Selhausen, the water-limited site (W-limited) and (B) Rollesbroich, the energy-limited site (E-limited) located in Eifel/Lower Rhine Valley (c) of the German Terrestrial Environmental Observatories (TERENO) in Germany. The $\Delta$Temp and $\Delta$P are the changes in temperature and precipitation. Modified from Rahmati et al. [37].

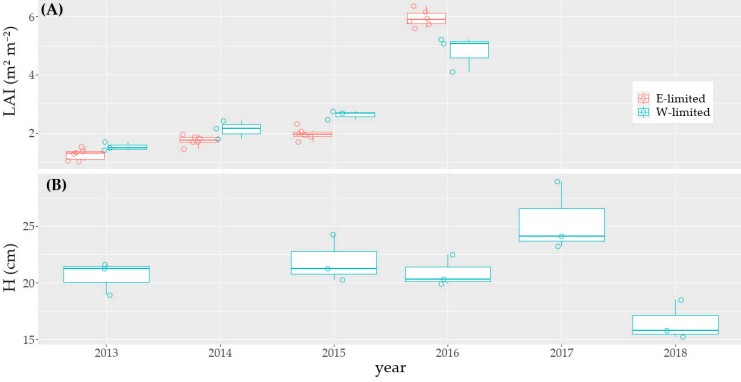

**Figure A2.** Mean annual grass height (H) for Rollesbroich, the energy-limited site (E-limited, blue) and Selhausen, water-limited site (W-limited, orange).

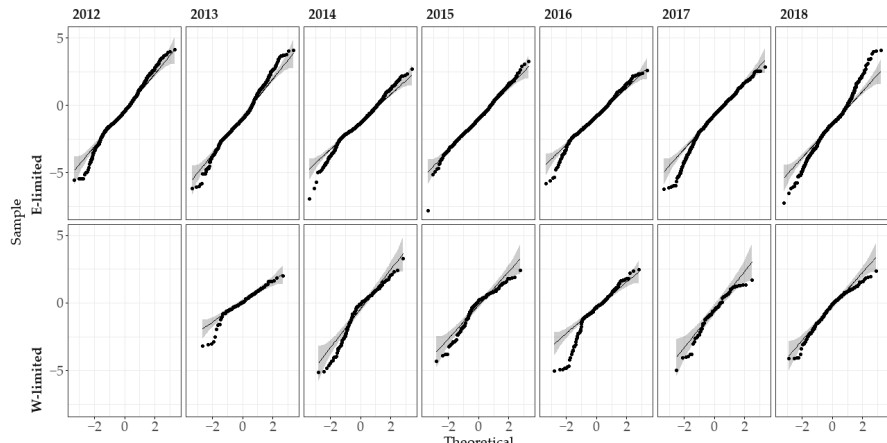

**Figure A3.** Quantile-quantile (Q-Q) plot comparing for each year and study site the distribution of the gross drained nitrogen (Sample) with a normal distribution (Theoretical). The black line represent a perfect correlation. The grey ribbon represents the 95% confidence level interval.

**Table A1.** Fertilizer application dates for Rollesbroich soil in Rollesbroich, the energy-limited site (E-limited) and Selhausen, the water-limited site (W-limited). "∅" indicates that no fertilization occurred.

|  | **E-Limited** | | | **W-Limited** | | |
|---|---|---|---|---|---|---|
| **Year** | **N°1** | **N°2** | **N°3** | **N°1** | **N°2** | **N°3** |
|  | **Day Month** | | | | | |
| 2012 | ∅ | ∅ | ∅ | ∅ | ∅ | ∅ |
| 2013 | ∅ | ∅ | ∅ | ∅ | ∅ | ∅ |
| 2014 | 24/03 | 24/11 * | ∅ | 27/03 | 24/11 * | ∅ |
| 2015 | 09/03 * | 08/07 | 18/11 * | 10/03 * | 08/07 | 18/11 |
| 2016 | 12/04 | 23/11 * | ∅ | 12/03 | 23/11 * | ∅ |
| 2017 | 21/02 * | 03/07 | ∅ | 21/03 * | 04/07 | ∅ |
| 2018 | 20/11 * | ∅ | ∅ | 20/03 * | ∅ | ∅ |

\* Applications done during the non-growing period of the grass.

**Table A2.** Components of the lysimeter annual soil water balance—mean values and standard deviations, $\pm$, in mm year$^{-1}$—for Rollesbroich soil in Rollesbroich, the energy-limited site (E-limited) between 2012 and 2018 with the precipitation (Prec), actual evapotranspiration (ETa), crop-height-adjusted reference evapotranspiration (ETcrop), drainage (Dr), capillary rise of water (CR), and change in soil water storage (ΔWS); negative values represent a loss of water for the system; row "mean" gives the means of the annual values for the period 2012–2018; the inter-annual variability is represented by the standard error of the annual mean values.

|  | **E-Limited** | | | | | |
|---|---|---|---|---|---|---|
| **Year** | **Prec** | **ETa** | **ETcrop** | **Dr** | **CR** | **ΔWS** |
|  | **mm year$^{-1}$** | | | | | |
| 2012 | 1035.7 ± 0.0 | −605.2 ± 23.5 | −648.0 ± 4.5 | −482.1 ± 78.3 | 15.2 ± 6.1 | −36.5 ± 59.0 |
| 2013 | 982.3 ± 11.8 | −652.5 ± 25.0 | −629.1 ± 15.4 | −451.1 ± 27.2 | 36.3 ± 17.3 | −85.0 ± 18.9 |
| 2014 | 1098.2 ± 11.2 | −685.1 ± 33.8 | −734.2 ± 31.5 | −425.5 ± 27.9 | 27.2 ± 13.6 | 14.9 ± 20.2 |
| 2015 | 1148.7 ± 15.5 | −675.0 ± 20.4 | −736.8 ± 17.6 | −567.6 ± 45.1 | 46.6 ± 15.2 | −47.3 ± 47.1 |
| 2016 | 1102.7 ± 15.4 | −617.5 ± 16.8 | −702.2 ± 19.7 | −529.3 ± 18.6 | 40.6 ± 16.6 | −3.6 ± 21.4 |
| 2017 | 1133.2 ± 18.6 | −659.8 ± 26.7 | −738.6 ± 14 | −541.2 ± 25.0 | 46.0 ± 18.7 | −21.9 ± 21.0 |
| 2018 | 917.6 ± 11.1 | −659.6 ± 30.5 | −804.5 ± 19.7 | −379.0 ± 34.0 | 53.0 ± 20.0 | −68.0 ± 19.1 |
| mean | 1059.8 ± 32.1 | −650.7 ± 11.0 | −713.3 ± 22.6 | −482.3 ± 25.7 | 37.8 ± 4.9 | −35.3 ± 13.3 |

**Table A3.** Components of the lysimeter annual soil water balance—mean values and standard deviations, $\pm$, in mm year$^{-1}$—for Rollesbroich soil in Selhausen, the water-limited site (W-limited) between 2012 and 2018 with the precipitation (Prec), actual evapotranspiration (ETa), crop-height-adjusted reference evapotranspiration (ETcrop), drainage (Dr), capillary rise of water (CR), and change in soil water storage ($\Delta$WS); negative values represent a loss of water for the system; row "mean" gives the means of the annual values for the period 2012–2018; the inter-annual variability is represented by the standard error of the annual mean values.

| | | | W-Limited | | | |
|---|---|---|---|---|---|---|
| Year | Prec | ETa | ETcrop | Dr | CR | $\Delta$WS |
| | | | mm year$^{-1}$ | | | |
| 2012 | 722.5 ± 0.0 | −689.9 ± 10.8 | −698.7 ± 11.2 | −13.6 ± 16.6 | 48.6 ± 8.2 | 67.7 ± 11.8 |
| 2013 | 588.0 ± 0.0 | −731.0 ± 20.3 | −793.4 ± 11.7 | −86.6 ± 7.7 | 108.8 ± 3.7 | −120.8 ± 11.3 |
| 2014 | 710.9 ± 0.0 | −736.8 ± 34.1 | −794.4 ± 35.9 | −64.1 ± 28.4 | 152.2 ± 66.7 | 62.2 ± 70.4 |
| 2015 | 749.1 ± 13.4 | −795.7 ± 15.0 | −860.2 ± 21.6 | −53.1 ± 13.2 | 82.1 ± 24.5 | −17.7 ± 35.5 |
| 2016 | 713.6 ± 8.4 | −657.2 ± 38.8 | −798.0 ± 9.2 | −56.9 ± 41.4 | 45.5 ± 29.1 | 45.0 ± 27.0 |
| 2017 | 648.9 ± 11.5 | −683.4 ± 26.8 | −850.9 ± 32.8 | −52.0 ± 36.5 | 127.2 ± 7.0 | 40.7 ± 29.4 |
| 2018 | 523.4 ± 8.7 | −586.8 ± 15.8 | −909.2 ± 20.7 | −67.1 ± 26.3 | 87.2 ± 16.4 | −43.3 ± 6.9 |
| mean | 665.2 ± 31.3 | −697.3 ± 25.1 | −815.0 ± 25.3 | −56.2 ± 8.4 | 93.1 ± 14.9 | 4.8 ± 26.2 |

**Table A4.** Components of the lysimeter soil nitrogen balance—mean values and standard deviations, $\pm$, in kg ha$^{-1}$ year$^{-1}$—for Rollesbroich soil in Rollesbroich, the energy-limited site (E-limited) between 2012 and 2018 with the wet deposition (N_WD), applied fertilizer (N_Fert), volatilization of the fertilizer nitrogen (N_Vol), plant nitrogen uptake (N_PU), nitrogen in drainage (N_Dr), nitrogen in capillary rise (N_CR), gaseous nitrous oxide nitrogen (N_N$_2$O), and change in soil nitrogen storage ($\Delta$N_S); negative values represent a loss of nitrogen for the system; column "mean" indicates the means of the annual values for the period 2012–2018 and the inter-annual variability is represented by the standard error of annual means. NA stands for Not Available.

| | | | | E-Limited | | | | |
|---|---|---|---|---|---|---|---|---|
| Year | N_WD | N_Fert | N_Vol | N_PU | N_Dr | N_CR | N_N2O | $\Delta$N_S |
| | | | | kg ha$^{-1}$ year$^{-1}$ | | | | |
| 2012 | 11.4 ± 0.0 | 0.0 ± 0.0 | 0.0 ± 0.0 | −171.4 ± 30.3 | −4.5 ± 6.6 | 0.0 ± 0.0 | NA ± NA * | −164.5 ± 35.5 |
| 2013 | 13.2 ± 0.2 | 0.0 ± 0.0 | 0.0 ± 0.0 | −174.1 ± 63.5 | −3.7 ± 5.4 | 0.0 ± 0.0 | −0.1 ± 0.1 * | −164.6 ± 67.7 |
| 2014 | 14.3 ± 0.1 | 65.3 ± 3.1 | −34.1 ± 1.3 | −195.1 ± 36.6 | −1.5 ± 0.7 | 0.0 ± 0.0 | −0.6 ± 0.3 | −151.3 ± 37.8 |
| 2015 | 11.9 ± 0.0 | 88.9 ± 5.8 | −62.2 ± 2.5 | −181.9 ± 28.0 | −1.7 ± 0.9 | 0.0 ± 0.0 | −0.8 ± 0.4 | −145.3 ± 29.5 |
| 2016 | 15.8 ± 0.2 | 63.5 ± 5.7 | −39.3 ± 2.4 | −177.9 ± 24.8 | −2.0 ± 0.6 | 0.0 ± 0.0 | −1.0 ± 0.9 | −140.4 ± 23.9 |
| 2017 | 14.4 ± 0.2 | 64.4 ± 1.6 | −39.8 ± 0.6 | −110.7 ± 29.6 | −2.0 ± 0.8 | 0.1 ± 0.1 | −0.7 ± 0.5 | −73.9 ± 29.1 |
| 2018 | 10.1 ± 0.1 | 45.4 ± 0.0 | 0.0 ± 0.0 | −117.0 ± 23.9 | −1.7 ± 1.1 | 0.1 ± 0.1 | −0.3 ± 0.2 * | −63.2 ± 23.3 |
| mean | 13.0 ± 0.8 | 46.8 ± 13.0 | −25.1 ± 9.5 | −161.2 ± 12.6 | −2.4 ± 0.4 | 0.0 ± 0.0 | −0.6 ± 0.1 | −129.0 ± 16.0 |

\* the N$_2$O emissions could only be computed between 07/2013 and 12/2018.

**Table A5.** Components of the lysimeter soil nitrogen balance—mean values and standard deviations, $\pm$, in kg ha$^{-1}$ year$^{-1}$—for Rollesbroich soil in Selhausen, the water-limited site (W-limited) between 2012 and 2018 with the wet deposition (N_WD), applied fertilizer (N_Fert), volatilization of the fertilizer nitrogen (N_Vol), plant nitrogen uptake (N_PU), nitrogen in drainage (N_Dr), nitrogen in capillary rise (N_CR), gaseous nitrous oxide nitrogen (N_N$_2$O), and change in soil nitrogen storage ($\Delta$N_S); negative values represent a loss of nitrogen for the system; column "mean" indicates the means of the annual values for the period 2012–2018 and the inter-annual variability is represented by the standard error of annual means. NA stands for Not Available.

| | | | | W-Limited | | | | |
|---|---|---|---|---|---|---|---|---|
| Year | N_WD | N_Fert | N_Vol | N_PU | N_Dr | N_CR | N_N2O | $\Delta$N_S |
| | | | | kg ha$^{-1}$ year$^{-1}$ | | | | |
| 2012 | 13.7 ± 0.0 | 0.0 ±0.0 | 0.0 ± 0.0 | −129.3 ± 24.6 | 0.0 ± 0.0 | 0.0 ± 0.0 | NA ± NA * | −115.6 ± 24.6 |
| 2013 | 9.9 ± 0.0 | 0.0 ± 0.0 | 0.0 ± 0.0 | −81.9 ± 6.6 | −0.6 ± 0.6 | 0.0 ± 0.0 | −0.3 ± NA * | −72.7 ± 6.1 |
| 2014 | 11.9 ± 0.0 | 55.9 ± 1.7 | −26.9 ± 0.6 | −187.1 ± 25.4 | −1.0 ± 0.7 | 0.0 ± 0.0 | −2.3 ± 0.3 | −149.4 ± 25.0 |
| 2015 | 10.2 ± 0.2 | 86.8 ± 1.8 | −64.2 ± 0.8 | −194.2 ± 2.5 | −0.9 ± 0.4 | 0.0 ± 0.0 | −1.2 ± 0.3 | −163.5 ± 2.9 |
| 2016 | 13.8 ± 0.4 | 66.1 ± 0.4 | −40.0 ± 0.2 | −137.8 ± 28.6 | −0.8 ± 0.7 | 0.6 ± 0.5 | −0.9 ± 0.3 | −99.1 ± 28.5 |
| 2017 | 12.4 ± 0.2 | 62.4 ± 1.8 | −42.7 ± 0.6 | −96.6 ± 8.7 | −0.3 ± 0.2 | 2.8 ± 0.4 | −0.6 ± 0.5 | −62.5 ± 9.6 |
| 2018 | 11.4 ± 0.2 | 45.4 ± 0.0 | 0.0 ± 0.0 | −83.8 ± 5.6 | −1.2 ± 0.7 | 1.8 ± 0.3 | −0.3 ± 0.1 * | −26.6 ± 6.1 |
| mean | 11.9 ± 0.6 | 45.2 ± 12.6 | −24.8 ± 9.7 | −130.1 ± 17.6 | −0.7 ± 0.2 | 0.7 ± 0.4 | −1.9 ± 0.3 | −98.5 ± 18.4 |

\* the N$_2$O emissions could only be computed between 07/2013 and 12/2018.

**Table A6.** Plant nitrogen uptake per harvest—mean values and inter-annual standard errors, $\pm$, in kg ha$^{-1}$ year$^{-1}$—for Rollesbroich soil in Rollesbroich, the energy-limited site (E-limited) and Selhausen, the water-limited site (W-limited) between 2012 and 2018.

| Site | Harvest Number | | | |
| | N°1 | N°2 | N°3 | N°4 |
| --- | --- | --- | --- | --- |
| | kg ha$^{-1}$ year$^{-1}$ | | | |
| E-limited | 64.2 ± 3.7 | 38.9 ± 2.3 | 44.5 ± 3.6 | 29.4 ± 2.7 |
| W-limited | 67.4 ± 0.6 | 27.3 ± 1.6 | 23.0 ± 2.3 | 29.1 ± 0.9 |

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
