# Peer review of "Soil Nitrogen Dynamics in a Managed Temperate Grassland Under Changed Climatic Conditions"

_water, doi:10.3390/w13070931_

Round 1

Reviewer 1 Report

The study tried to clarify the impact of climate change, i.e., drying and heating, on N dynamics in grassland. I noticed that the dataset is very nice, and the paper deserves publication. However, the presentation must be improved much.

The introduction needs improvement. Impacts of drying and warming on N dynamics in soils are not very special in grassland. The present version of introduction depends on few studies in grassland, but there should be much more knowledge about the relationship between drying/warming and soil N dynamics. More information should be presented in the introduction section.

The authors did not make full use of their data. The rough analysis of just comparing yearly-average between the two treatments authors prepared (for example, Fig. 2, 4) lowers the value of large dataset of the authors. The conclusion or concept of the study is not novel, so the value of the paper should rely on the preciseness of the dataset. I recommend large revision throughout the paper.

The research question and methodology to resolve the question is not well arranged. In L84, authors mentioned, “The above literature review allows us to hypothesis that a change of the ecosystem from an energy- to a water-limited site with overall drier and warmer climatic conditions and altered seasonal Prec distribution will lead to a decreasing grassland N_PU since the grassland ETa will be limited by water stress [22,37].” But whether the grassland N uptake decreases or not depends on the strength of the drought, doesn’t it? In addition, L87 says, “This will contribute to an increase in net mineralisation rate [22].” It also depends on the strength of drought. If it is a really dry condition, mineralization must be reduced. Is it really possible to clarify the impact of drying and warming on N dynamics by just comparing two treatments? Authors should justify the consistency of the research question and their methodology.

I just could not understand the hypothesis written in L84. Light-limited for plants? Or carbon limited for heterotrophs?

Presentation of figures should be improved. Do not use the name “Ro-in-Se” or “Ro-in-Ro.” For readers it is not easy to understand. Rename using more general words describing the treatment.

It is confusing to compare N_PU using minus values in Fig. 4G. Plant N uptake is higher in Ro-in-Ro, right? Please make it clearer.

Reviewer 2 Report

Comments (Water Manuscript ID: water-1110453)

The manuscript entitled “Soil Nitrogen Dynamics in a Managed Temperate Grassland Under Changed Climatic Conditions” was intended to exam the uptake, minimization, leaching and gaseous emission processes of nitrogen in soil monoliths with grasses due to water resource limitation in the period between 2012 and 2018. The study found that plant uptake of nitrogen decreased under water resource decrease and correspondingly net mineralization rate increased. Based on the results, the authors suggested that fertiliser should be applied early in the growing period in the studied grasslands to increase nitrogen uptake and decrease nitrogen losses. The topic is interests and the results from this study provided useful data and information for better understanding of changes in nitrogen in temperate grasslands under climate changes. The content of the manuscript meet the scope of the journal ‘Water’ for publication. However, it seems to me that the scientific quality of this manuscript, especially in experimental design, does not satisfy the requirements of the journal for publication at this version. I would like to provide my comments in detail when the manuscript is revised.

  1. What is the concept of energy-limited site and water-limited site? As the two locations (Rollesbroich and Selhausen) are very close in the distance, what is the main reason cause the differences of climate variables between the two sites. Are they still with the similar range of energy and water recourse? It would be better if the authors could provide a map showing the locations of the study.
  2. What is the experimental design for this study? When the soil monoliths transported from Rollesbroich to Selhausen, do the authors consider the changes in grass growth and development conditions and soil organisms (species, abundance, composition, and activity) which will affect the depend variables (nitrogen dynamics), besides considering the changes in energy and water sources (as independent Variables).
  3. Based on the study, it seems that all soil monoliths in Rollesbroich should be as the ‘standard’ sites, and those in Selhausen as the ‘treated’ sites. How do the authors maintain all other conditions the similar between the two sites, except the difference of water recourse?
  4. As the study lasts 6-7 years (from 2012 and 2018), how does the time affect the nitrogen dynamics in the soil monoliths? How is the interactive effect of energy and water with time on nitrogen dynamics? It would be better if the authors could perform a statistic analysis for this interaction.
  5. As so many abbreviations are used in the manuscript, it would be better if the authors could develop a table to show the meanings of all abbreviations.
  6. Some contents in the Results section should be removed to the Discussion or Materials and Methods sections. It is no necessary for the authors to explain why some events occurred in the Results section. For example, Line 299-305, Line 380-390, Line 392-397, Line 400-401.
  7. It has been recognized that ‘The space-for-time substitution approach’ would introduce a large uncertain or errors when conduct researches in the fields of Hydrology and Ecology. Hopefully, we can see some sentences related to the uncertain or limitations due to use ‘The space-for-time substitution approach’ in the Discussion of this manuscript.
